



# 1 Tectonostratigraphic evolution of the Slyne Basin

Conor M. O'Sullivan [1,2,4], Conrad J. Childs [1,2], Muhammad M. Saqab [1,2,5], John J. Walsh [1,
2], Patrick M. Shannon [1,3]
[1] Irish Centre for Research in Applied Geoscience (iCRAG), O'Brien Centre for Science (East), University College
Dublin, Belfield, Dublin 4, Ireland
[2] Fault Analysis Group, School of Earth Sciences, University College Dublin, Belfield, Dublin 4, Ireland
[3] School of Earth Sciences, University College Dublin, Belfield, Dublin 4, Ireland
[4] Present address: Petroleum Experts, Petex House, 10 Logie Mill, Edinburgh, EH7 4HG, United Kingdom
[5] Present address: Norwegian Geotechnical Institute, 40 St. Georges Terrace, Perth, WA 6000, Australia
Corresponding author email: cmnosullivan@gmail.com

## 11 1. Abstract

The Slyne Basin, located offshore NW Ireland, is a narrow and elongate basin composed of a
series of interconnected grabens and half-grabens, separated by transfer zones coincident
with deep Caledonian-aged crustal structures. The basin is the product of a complex,
polyphase structural evolution stretching from the Permian to the Miocene. Relatively low-
strain episodic rifting occurred in the Late Permian and the latest Triassic to Middle Jurassic,
with the main phase of rifting occurring in the Late Jurassic. These extensional events were
punctuated by periods of tectonic quiescence during the Early Triassic, and regional uplift and
erosion during the late Middle Jurassic. Late Jurassic strain was primarily accommodated by
several kilometres of slip on the basin-bounding faults, which formed through the breaching of
relay ramps between left-stepping fault segments developed during earlier Permian and Early-
Mid Jurassic rift phases. Following the cessation of rifting at the end of the Jurassic, the area
experienced kilometre-scale uplift and erosion during the Early Cretaceous and second, less-
severe phase of denudation during the Palaeocene. These post-rift events formed a distinct
regional post-rift unconformity and resulted in a reduced post-rift sedimentary section. The
structural evolution of the Slyne Basin is influenced by pre-existing Caledonian structures at a
high angle to the basinal trend. The basin illustrates a rarely documented style of fault
reactivation in which basin-bounding faults are oblique to the earlier structural trend, but the
initial fault segments are parallel to this trend. The result is a reversal of the sense of stepping
of the initial fault segments generally associated with basement control on basin-bounding
faults.





## 2. Introduction

The north-western European Atlantic margin is made up of a framework of basins which are the product of a polyphase geological evolution stretching from Variscan orogenic collapse to the formation of oceanic crust during the opening of the North Atlantic Ocean (Fig. 1A). The evolution of these basins is influenced by a variety of factors, including pre-existing faults and lineaments, typically inherited from the Caledonian or Variscan orogenies, and the presence of salt within the sedimentary basin-fill, acting as layers of mechanical detachment. Pre-existing structures have been observed both reactivating during later rift events if oriented optimally (e.g. Stein, 1988; Schumacher, 2002; Wilson et al., 2010; Bird et al., 2014; Fazlikhani et al., 2017; Osagiede et al., 2020) or acting as barriers to fault growth and segmenting rift systems if they are oblique to the extension direction (e.g. Morley et al., 2004; Pereira et al., 2011; Philips et al., 2018).









**Figure 1: A)** Simplified structural map of the NW European Atlantic margin showing the study area in relation to other Permian & Mesozoic sedimentary basins, adapted from Doré et al., 1999 and Naylor et al., 1999. Caledonian structural lineaments which segment the basins are lighted in red. **Abbreviations:** GGFZ, Great Glen Fault Zone; HBFC, Highland Boundary- Fair Head–Clew Bay Lineament; KBB, Kish Bank Basin; SUAG, Southern Uplands- Antrim-Galway Lineament; UB, Ulster Basin; WOB, West Orkney Basin; WSB, West Shetland Basin.. **B)** Time structure map of the Base Permian or Variscan Unconformity in the Slyne Basin. Local sub-basins and structural elements are labelled. **Abbreviations:** CSTZ – Central Slyne Transfer Zone.

The Slyne Basin is a narrow chain of grabens and half-grabens that occupy the eastern margin of the Rockall Basin (Fig. 1). The Slyne Basin shows significant along-strike structural variability, with changes in dip direction of the kilometre-scale basin-bounding faults occurring over relatively short distances i.e. transfer zones. The transfer zones have been interpreted as areas where crustal-scale lineaments and terrane boundaries of Caledonian age transect the younger Late Palaeozoic and Mesozoic rifts. Similar phenomena have been observed in rift basins across the world, where pre-existing zones of weakness can be reactivated if oriented optimally. The north-western European Atlantic margin is underlain by a series of pre-existing structures and structural inheritance and reactivation has been well documented in the Norwegian and UK Atlantic margins (Doré et al., 1999; Doré et al., 2007; Ady & Whittaker, 2019; Schiffer et al., 2019) as well as in the Iberian Atlantic margin (Alves et al., 2006; Pereira et al., 2017).

The structural geology of the Slyne Basin was the subject of significant study during the late 1990s and early 2000s following the discovery of the Corrib gas field in 1996 (Dancer et al., 2005). Previous publications documented aspects of the structural evolution (Chapman et al., 1999; Dancer et al., 1999) and the role of exhumation in the petroleum system of the basin (Corocoran & Doré, 2002; Corocoran & Mecklenburgh, 2005), as well placing the basin in the regional context of the Irish Atlantic margin (e.g. Corfield et al., 1999; Walsh et al., 1999). In recent years, significantly more and higher quality seismic data, together with additional well data have been acquired throughout the Slyne Basin and neighbouring areas (Shannon, 2018). Additionally, a comprehensive biostratigraphic study of all the Irish offshore basins has reclassified the ages of key syn-rift sequences (Merlin Energy Resources Consortium, 2020), warranting fresh investigation into the structural evolution of the Slyne Basin and its context within the greater Irish Atlantic margin.

This study utilizes an extensive database of borehole-constrained 2D and 3D seismic reflection data, coupled with the results from the new biostratigraphic database, to investigate the structural evolution of the Slyne Basin. Key aspects of this structural history, including the development of the major basin-bounding faults, the role of salt in basin evolution, and influence of pre-existing crustal-structures in the segmentation of the Slyne basin are



examined and characterised. These findings are then placed in a regional context to better
understand the role of the Slyne Basin in the evolution of the greater Irish Atlantic margin.

## 3. Geological Setting

The Slyne Basin has a relatively flat present-day bathymetry, with water depths ranging from
100 to 600m across most of the study area, with water depths increasing up to 2500m in the
north (Dancer et al., 1999). It is divided into three distinct sub-basins: the Northern, Central
and Southern Slyne sub-basins (Fig. 1, sensu Trueblood & Morton, 1991). These are
separated by transfer zones (e.g. Morley et al., 1990; Gawthorpe & Hurst, 1993) which
coincide with the location of major structural lineaments, in the form of Caledonian terrane
boundaries.
The Slyne Basin is bounded along its eastern margin by the Irish Mainland Shelf, while the
Porcupine and Slyne highs make up the western boundary (Fig. 1B). The Colm Basin,
previously identified as a distinct Mesozoic basin (Dancer et al., 1999), appears to be an
extension of the Northern Slyne Sub-basin, verging south-westwards between the Rockall
Basin and the Porcupine High. A narrow, discontinuous basement horst which represents a
southern extension of the Erris Ridge (Cunningham & Shannon, 1997) separates the Northern
Slyne Sub-basin and the neighbouring Erris Basin from the Rockall Basin to the northwest.
Similarly, a narrow basement high separates the Southern Slyne Sub-basin from the
Porcupine Basin to the southwest.

### 3.1.    Basement configuration

Previous authors have noted the role of pre-existing Caledonian structures in the
segmentation of younger Mesozoic basins on the Irish Atlantic margin, correlating the offshore
extension of these crustal-scale structures with complex transfer zones separating distinct
sub-basins (Trueblood & Morton, 1991; Dancer et al., 1999). Several authors have mapped
the offshore extent of Caledonian structural lineaments on the Irish Atlantic margin (Lefort &
Max, 1984; Tate, 1992; Naylor & Shannon, 2005; Štolfová & Shannon, 2009; Kimbell et al.,
2010). There are three Caledonian structures relevant to the evolution of the Slyne Basin; the
Great Glen Fault Zone (GGFZ), the Highland Boundary-Fair Head Clew Bay Fault Zone
(HBFC) and the Southern Uplands-Antrim Galway Fault Zone (SUAG). The exact locations of
these structures in the vicinity of the Slyne Basin are variably constrained; the NE-SW trending
GGFZ has been mapped across the Irish Mainland Shelf to the west of the Erris Basin using
deep seismic profiles and potential field datasets as a vertical strike-slip fault (Klemperer et
al., 1991; Kimbell et al., 2010). The GGFZ intersects the Slyne Basin between the Northern
and Central Slyne sub-basins at a location termed the Central Slyne Transfer Zone (CSTZ,





sensu Dancer et al., 1999). The HBFC and SUAG structures are more poorly constrained; the
HBFC is an E-W oriented structure bounding the southern shore of Clew Bay on the west
coast of Ireland and is mapped passing through Clare Island due west of Clew Bay (Fig. 1,
Badley, 2001; Worthington & Walsh, 2011). The HBFC may correlate with the fault zone
separating the Central and Southern Slyne sub-basins, but there is also evidence that splays
of the HBFC may also be observed in the Central Slyne Sub-basin (Fig. 1B). The SUAG
structure has been mapped trending E-W along the northern shore of Galway Bay (REF) and
south of the Slyne Basin, through the Brendan Igneous Centre (Fig. 1). These lineaments
separate different basement terranes which were assembled during the Caledonian Orogeny
and have been extended from their known extents onshore Ireland and Scotland by several
authors (e.g. Roberts et al., 1999; Tyrrell et al., 2007; Štolfová & Shannon, 2009). Limited pre-
Carboniferous well penetrations in the Slyne Basin preclude the accurate mapping of these
basement terranes and the interpretations of previous authors are adopted here.

## 3.2.    Stratigraphic framework of the Slyne Basin

Previous stratigraphic nomenclature for the Slyne Basin was largely based on comparisons
with the geology of the Hebridean basins exposed on the Isle of Skye (e.g. Trueblood &
Morton; 1991). An updated stratigraphic nomenclature with revised biostratigraphy has
recently been published, standardising nomenclature at group, formation and member levels
across the sedimentary basins of the Irish Continental Shelf (Merlin Energy Resources
Consortium, 2020). This stratigraphic nomenclature is used in this study (Fig. 2). The main
Middle Jurassic syn-rift section of previous authors (e.g. Chapman et al., 1999; Dancer et al.,
1999; Corcoran & Mecklenburgh, 2005; Dancer et al., 2005) has recently been reclassified as
Late Jurassic in age. This has important implications for regional geodynamics which are
discussed below. For full details on the biostratigraphic reclassification please refer to Merlin
Energy Resources Consortium (2020).



**Figure 2:** Lithostratigraphic chart showing the simplified stratigraphy of the Slyne Basin, age of key horizons, and typical seismic stratigraphy from the Central Slyne Sub-Basin. Adapted from Merlin Energy Resources Consortium, 2020.

The pre-rift section of the Slyne Basin consists of Carboniferous mudstones and sandstones with interbedded layers of coal, underlain by Silurian and older metasediments (Merlin Energy Resources Consortium, 2020). This is overlain by an evaporite-rich Permian sequence which is a lateral equivalent of the Zechstein Group encountered throughout NW Europe (O'Sullivan



et al., 2021). The overlying Triassic section is sub-divided into Lower Triassic fluvial
sandstones and Upper Triassic playa, sabkha and lacustrine mudstones and claystones
interbedded with thin limestone, sandstone, and anhydrite beds (Dancer et al., 2005; Štolfová
& Shannon, 2009). A massive halite unit is present at the base of the Upper Triassic section
in the northern Slyne Basin but is absent in the central and southern parts of the basin
(O'Sullivan et al., 2021).
The Lower Jurassic section is composed of marine sandstone, mudstones and carbonates,
overlain by Middle Jurassic calcareous marine mudstones (Trueblood & Morton, 1991; Dancer
et al., 1999). The Kingfisher Limestone Member is a unit of thick limestones that occurs at the
base of Kestrel Formation (sensu Merlin Energy Resources Consortium, 2020) and which
forms a distinct, semi-regional seismic marker termed the 'Bajocian Limestone Marker' in
previous literature (e.g. Trueblood & Morton, 1991; Scotchman & Thomas, 1995; Dancer et
al., 1999). A regional unconformity separates the underlying Lower and Middle Jurassic
sections from the overlying Upper Jurassic sediments. The Upper Jurassic section consists of
terrestrial and fluvio-estuarine mudstones and sandstones with numerous palaeosols and coal
layers, which are overlain by the marine mudstones, indicating a regional marine transgression
occurred during the late Oxfordian to Tithonian (Merlin Energy Resources Consortium, 2020).
The Base-Cretaceous Unconformity separates the Cretaceous section of the Slyne Basin from
the underlying Jurassic strata. The Lower Cretaceous stratigraphy consists of Albian-aged
glauconitic mudstones and sandstones, while the overlying Upper Cretaceous section is
composed of limestones and calcareous mudstones.  The Base-Cenozoic Unconformity forms
the lower boundary to the Cenozoic succession in the Slyne Basin. The Cenozoic section can
be subdivided into three sequences: a layer of Eocene lava locally developed in the northern
and southern areas of the Slyne Basin, overlain by an Eocene-Miocene section and a Miocene
to Quaternary section, both consisting of poorly consolidated marine mudstones and
sandstones, separated by a mid-Miocene unconformity.





## 4. Dataset & Methodology

This study focused on the interpretation of an extensive suite of multi-vintage 2D and 3D seismic reflection data collected during hydrocarbon exploration in the Slyne Basin (Fig. 3). The 2D seismic dataset consists of 17 surveys acquired between 1980 and 2007, comprising over 22,000 line-kilometres of data, while the 3D seismic dataset consists of eight surveys acquired between 1997 and 2013 and covers almost 4,000 square-kilometres. Seismic data quality varies from very poor to good, with the more modern vintages typically providing clearer imaging. Data quality in the Slyne Basin is heavily influenced by the near-seabed geology, with the distribution of Cenozoic lava flows and intrusive sills, coupled with Cretaceous chalk causing imaging problems including multiples, energy scattering and signal attenuation (Dancer & Pillar, 2001). These problems are most severe in the Northern Slyne Sub-basin, and the western margin of the Southern Slyne Sub-basin. The application of modern processing techniques and use of 3D seismic data has improved data quality in the region somewhat (Dancer & Pillar, 2001; Droujinine et al., 2005; Rohrman, 2007; Hardy et al., 2010), most recently with the acquisition of an ocean-bottom cable survey over the Corrib gas field in 2012 and 2013 (Shannon, 2018). Seismic sections are presented in European polarity (Brown, 2001), where a positive downwards increase in acoustic impedance corresponds to a positive (red) reflection event and a decrease corresponds to a negative (blue) reflection event. All sections are vertically exaggerated by a factor of three and ball-ends are used to highlight where a fault terminates within a certain stratigraphic package, while faults without ball-ends are truncated by a younger unconformity.



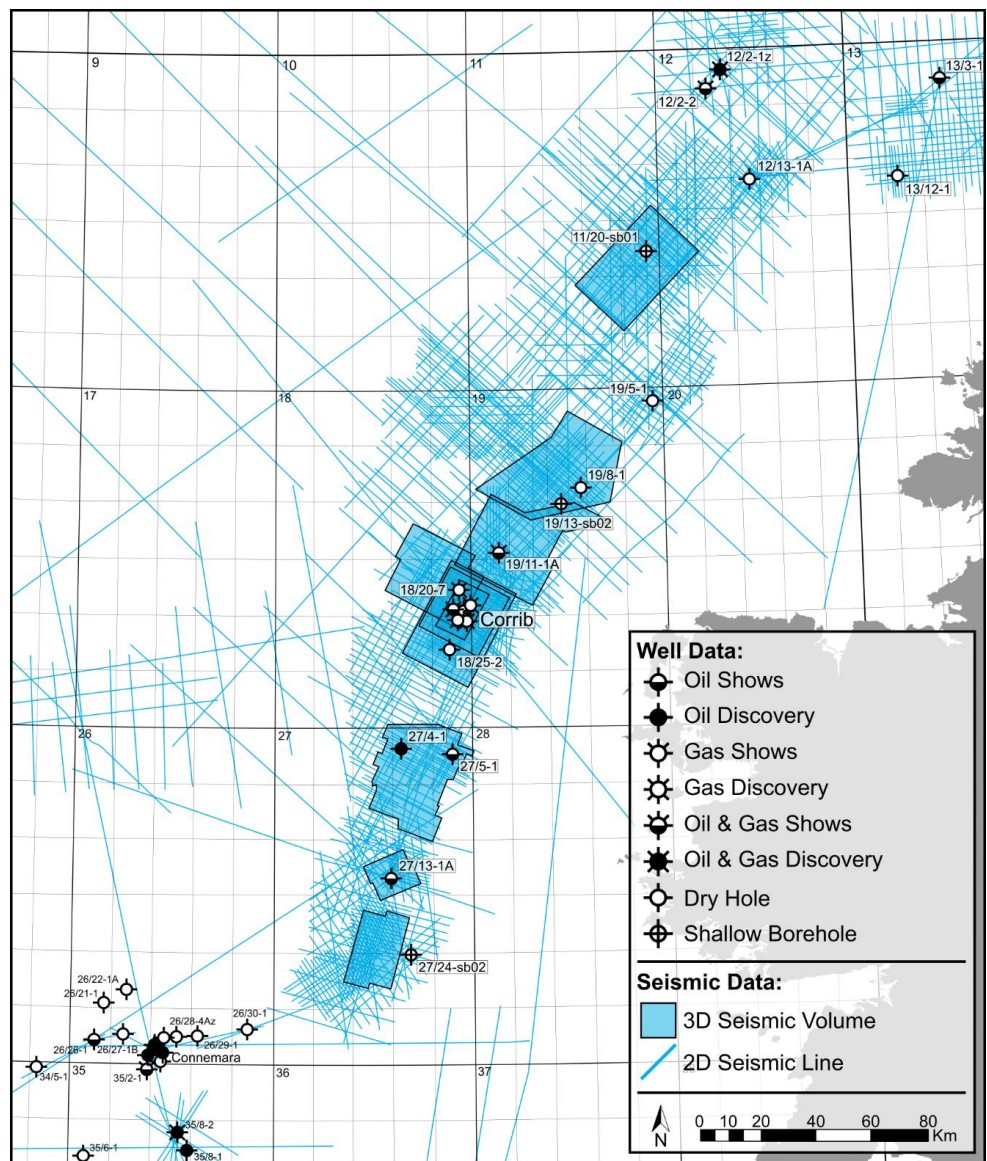

**Figure 3:** Map showing study area and data sets used.

Thirteen key horizons were mapped across the Slyne Basin in the time domain (Fig. 2). The ages of these horizons were constrained using exploration and appraisal wells in addition to shallow boreholes. The Northern Slyne Sub-basin has the highest well density, including eight appraisal and production wells associated with the Corrib gas field, and four near-field exploration wells (19/8-1, 19/11-1A, 18/20-7 and 18/25-2), with a further three exploration wells in the Central Slyne Sub-basin (27/4-1, 27/5-1 and 27/13-1). The stratigraphy of the Southern Slyne Sub-basin is unconstrained except for a single shallow borehole (27/24-





sb02A) which proved Lower Jurassic and Upper Triassic sediments beneath the Base-
Cenozoic Unconformity (Fugro, 1994a). The dataset associated with the exploration, appraisal
and production wells consist of comprehensive suites of wireline logs (gamma, caliper,
neutron-density, sonic, and resistivity logs), well completion reports with formation tops, and
time-depth relationship data as either checkshots, or vertical seismic profiles (VSPs).

## 212   5. Results

### 213    5.1.     Basin geometry & transfer zones

The Southern and Central sub-basins are half-grabens which dip towards the northwest (Fig.
4 & 5), with a NNE-SSW oriented basin-bounding fault separating them from the Porcupine
High to the west. As no Permian or Mesozoic strata are preserved on the footwall of these
basin-bounding faults (the Porcupine High) either through non-deposition or erosion (Fig. 4 &
5), the total throw on these faults is difficult to constrain. Nevertheless, the elevation of the
Base-Permian Unconformity in the adjacent hanging wall provides a minimum throw estimate
of 3000 ms TWTT (two-way travel time) along most of the length of this fault (Fig. 1B). Unlike
its Southern and Central neighbours, the Northern Slyne Sub-basin is an eastward-dipping
graben (Fig. 6 & 7) bounded by a series of segmented faults along its eastern boundary with
the Irish Mainland Shelf (Fig. 1B), while a narrow basement horst separates it from the Rockall
Basin to the NW. The fault system bounding the eastern margin of the Northern Slyne Sub-
basin consists of a series of left-stepping, NE-SW oriented faults linked by relay ramps (Fig.
1B). These faults are of a similar scale to the fault bounding the Southern and Central sub-
basins, with over 3000 ms TWTT of throw recorded (Fig. 1B). The northernmost segment of
this fault system separates the Slyne Basin from the Erris Basin to the north, with the Erris
Basin being downthrown relative to the Northern Slyne Sub-basin across this fault (Fig. 8).





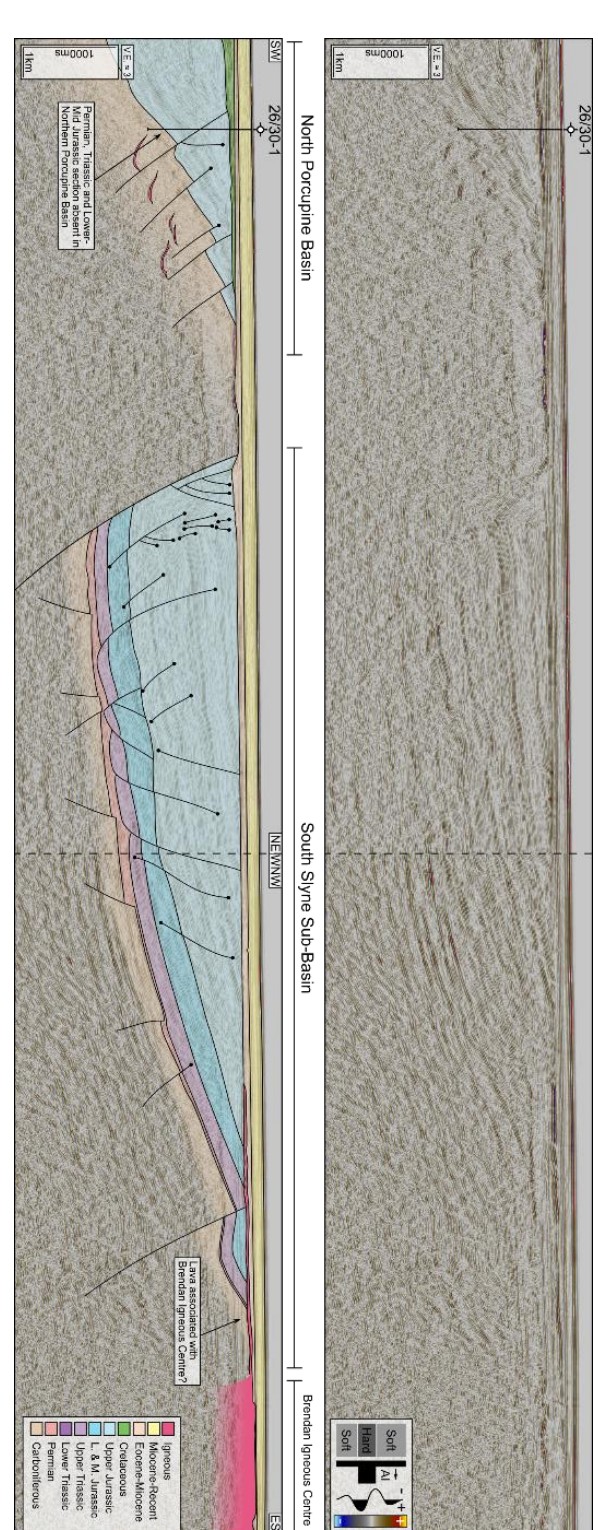




**Figure 4:** Composite section of 2D seismic lines NWI-93-202 and NWI-93-028 and
accompanying geoseismic interpretation covering the Southern Slyne sub-basin, North
Porcupine Basin, and Brendan Igneous Centre. The Southern sub-basin is a westward-
dipping half-graben, and is downthrown relative to the North Porcupine Basin, separated by a
narrow high composed of crystalline basement. See Figure 1 for location.





**Figure 5:** Composite seismic section of 2D seismic line E96IE09- 28 and inline 2740 from the 2000/08 (E00IE09) 3D seismic volume from the Central Slyne Sub-basin, with accompanying seismic interpretation. See Figure 1 for location. **Abbreviations:** PH – Porcupine High.




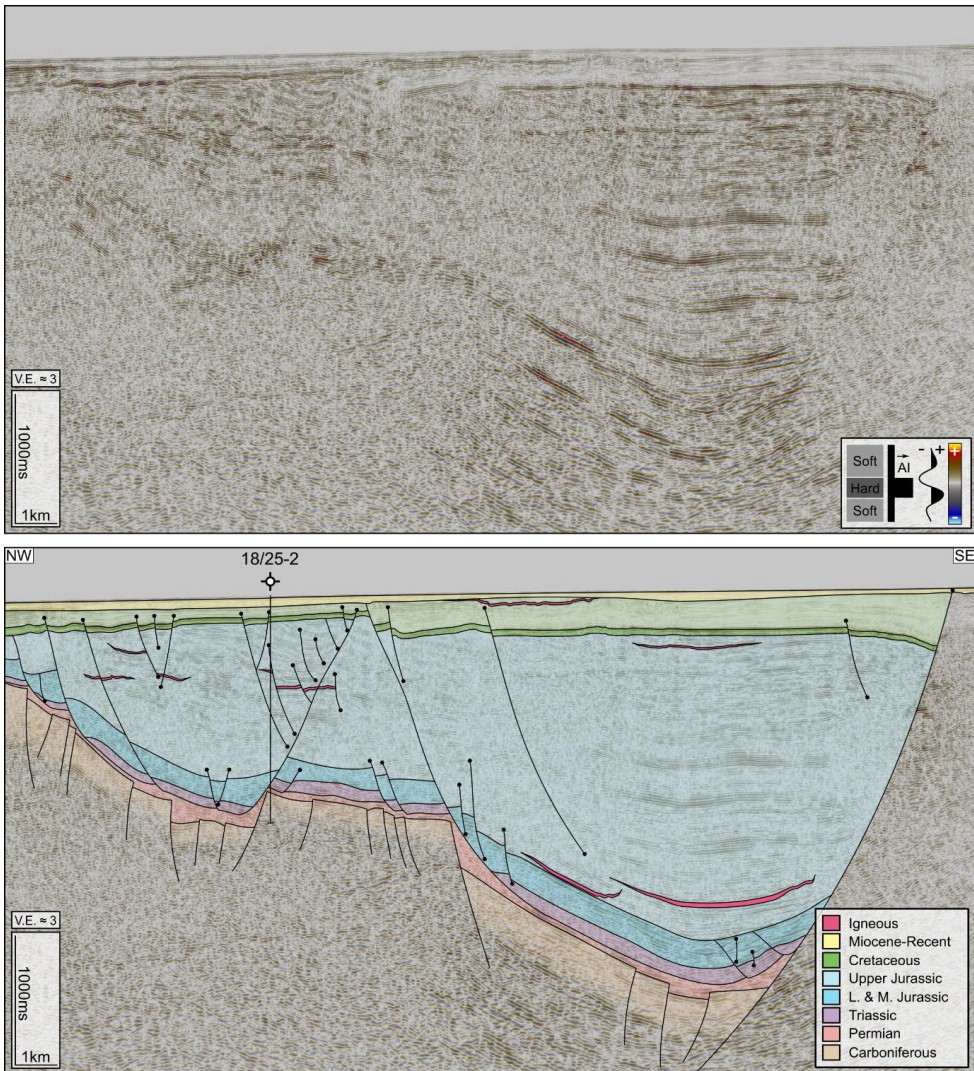

**Figure 6:** 2D seismic line E93IE07 and accompanying geoseismic interpretation from the Central Slyne Transfer Zone. Basin polarity has switched from the westward-dipping half-graben geometry of the Central and Southern sub-basins to an eastward-dipping half-graben geometry. The presence of near-seabed Upper Cretaceous Chalk causes a significant reduction in image quality. See Figure 1 for location.









**Figure 7:** Composite section of an arbitrary line from the Iniskea 2018 3D volume and 2D
seismic line ST9808-1002 from the Northern Slyne sub-basin, and accompanying geoseismic
interpretation. Significantly thicker Zechstein salt in this part of the Slyne Basin forms salt-
pillows and salt-anticlines, folding the overlying Mesozoic section. Detachment on the Uilleann
Halite causes rafting and listric faulting in the overlying Jurassic section. See Figure 1 for
location. **Abbreviations:** ER – Erris Ridge.



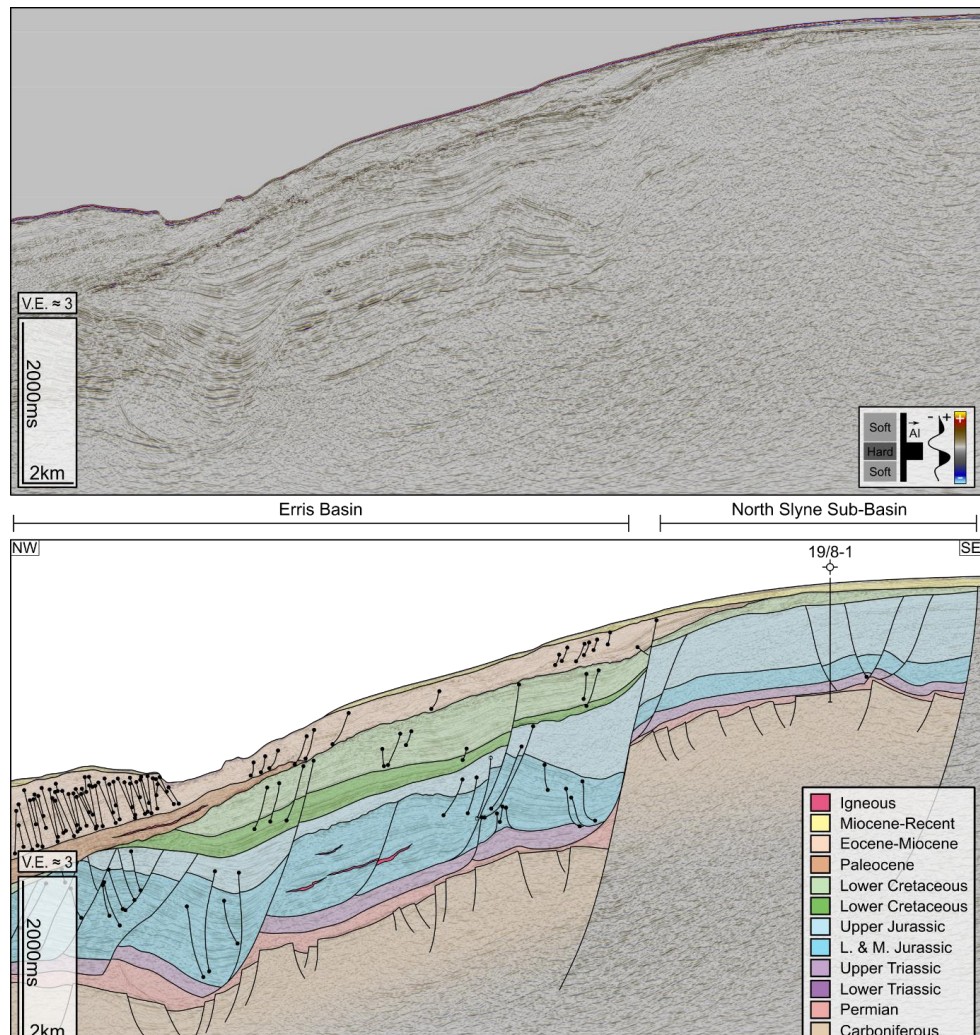


**Figure 8:** 2D seismic line ST9505-430 and accompanying geoseismic interpretation covering the Northern Slyne sub-basin and the southern Erris Basin. The Erris Basin is downthrown relative to the Slyne Basin, and has a significantly thicker Lower and Middle Jurassic section preserved, but conversely reduced Upper Jurassic stratigraphy. Significantly thicker Cretaceous and Cenozoic post-rift stratigraphy is preserved in the Erris Basin relative to the Slyne Basin. See Figure 1 for seismic line location.

The reversal of basin polarity occurs across the CSTZ, which coincides with the intersection of the offshore extension of the GGFZ and the Slyne Basin. Deep seismic transects adjacent to the Slyne Basin image the GGFZ as a NE-SW trending vertical discontinuity which appears to offset the Moho (Klemperer et al., 1991). The throw on the basin-bounding faults north and south of the CSTZ rapidly decreases as they approach the CSTZ so that horizons are continuous between the basins and strain is transferred between the faults of opposed polarity





via a convergent, conjugate transfer zone (sensu Morley et al., 1990). Both faults have over
3000 ms TWTT of throw on the Base Permian Unconformity within 10 kilometres of the CSTZ
(Fig. 1, 5, 6), with this value likely being an underrepresentation of the true throw given the
kilometre-scale erosion of Jurassic sediments recorded both north and south of the CSTZ
beneath the post-rift unconformities (e.g. Corcoran & Mecklenburgh, 2005; Biancotto et al.,
2007). In addition to the faults bounding the Central and Northern Slyne sub-basins, a NE-SW
oriented, southward dipping fault bounds the Slyne Embayment, a small half-graben to the
southwest of the CSTZ (Fig. 1B, 5). This suggests that the GGFZ acted as a barrier to the
propagation of the basin-bounding fault systems to both the north and south. The GGFZ is
likely linked to both the fault bounding the Slyne Embayment and the southernmost segment
of fault system bounding the Northern Slyne Sub-basin, both of which are subparallel to this
major regional structure.
The HBFC fault is interpreted as a hard-linked NE-SW oriented fault, dipping towards the NW,
which downthrows the Central Slyne Sub-basin relative to the Southern Slyne Sub-basin (Fig.
1B). The HBFC fault also appears to offset the NNE-SSW oriented fault bounding the Central
and Southern Slyne sub-basins (Fig. 1B), which may be a product of both normal dip-slip
movement observed offshore on seismic data and strike-slip movement recorded onshore
Ireland (e.g. Worthington & Walsh, 2011; Anderson et al., 2018). The nature of the interaction
between these two faults is unclear due to poor seismic image quality caused by shallow
Cenozoic lavas which blanket the western margin of the Southern Slyne Sub-basin. However,
the lateral offset of the NE-SW oriented basin-bounding fault and the adjacent Porcupine High
either side of the FHCB fault is well imaged on seismic sections immediately north and south
of this zone.

## 5.2.    The role of salt in basin development

The Slyne Basin contains two layers of salt; the Permian Zechstein Group and the Upper
Triassic Uilleann Halite Member (Fig. 2; Dancer et al., 2005; Merlin Energy Resources
Consortium, 2020; Fig. 2). The Zechstein Group is composed predominately of halite and
gypsum, while the Uilleann Halite Member is composed predominantly of halite interbedded
with red mudstone and anhydrite (O'Sullivan et al., 2021).
In the Central and Southern sub-basins, south of the CSTZ, only the Zechstein Group salt is
present (Fig. 2), where it mechanically detaches the sub-salt basement from the Mesozoic
supra-salt basin-fill (Fig. 4-5, 9, 10). Several halokinetic structures are present in the Central
and Southern Slyne sub-basins, including large salt rollers, collapsed diapirs and rafted fault
blocks (Fig. 5, 9). There are also several high-relief monoclines adjacent to the basin-bounding
fault in the Central Slyne Sub-basin which have been noted by previous authors (Fig. 5;



Dancer et al., 1999). The Triassic and Lower-Middle Jurassic section in these structures is
encountered at a similar depth to the same section along the eastern margin of the basin, and
the Triassic section appears to have welded to the crystalline basement of the Porcupine High
across the fault plane of the basin-bounding fault (Fig. 5). These structures likely formed
initially as forced folds above the sub-salt basin-bounding faults during the early stages of
rifting in the Late Jurassic, resulting in the Upper Jurassic section onlapping the flank of these
structures. As extension continued the fault breached the salt and led to the present geometry
(O'Sullivan et al., 2021).












**Figure 9:** Seismic sections and accompany interpretations showing salt structures in the Slyne Basin. **A)** Map showing the distribution of Upper Triassic and Permian salt in the Slyne Basin. Adapted from O'Sullivan et al., 2021. **B)** Seismic and geoseismic section through the Corrib gas field showing the kinematic interaction between Upper Triassic and Permian salt. The Permian salt forms a NE-SW oriented salt pillow, while the Upper Triassic forms an elongate salt wall parallel to the fold-axis of the salt pillow. Adapted from O'Sullivan & Childs, 2021. **C)** Several salt-related structures in the Central Slyne Sub-basin, including a salt pillow, salt roller and an apparent collapsed salt diapir. **D)** A large salt roller from the Southern Slyne Sub-basin. The fault in the supra-salt section appears to have hard-linked with the sub-salt basement fault.

In the Northern sub-basin both the Permian and Upper Triassic salt layers are present (Fig. 9A; Corcoran & Mecklenburgh, 2005; O'Sullivan et al., 2021). Here, both layers mechanically detach the stratigraphy above and below them, with the Permian salt detaching the Lower Triassic from the Carboniferous basement, while the Upper Triassic salt detaches the Jurassic section from the Lower Triassic (Fig. 7, 9B). Halokinetic structures formed in the Permian and Triassic salts are often coincident and can be demonstrated to be kinematically related. This is exemplified by the structure containing the Corrib gas field (Fig. 7, 9; Corcoran & Mecklenburgh, 2005; Dancer et al., 2005); here, the Permian salt forms a NE-SW trending salt pillow, which folds the overlying Mesozoic sediments. An Upper Triassic salt wall formed parallel to the fold-axis of the Permian salt pillow and forms the footwall to a listric delamination fault which downthrows the folded Jurassic section to the SE (Fig. 7, 9B). The evolution of the Corrib gas field is discussed in detail in O'Sullivan & Childs (2021).

Several of the halokinetic structures in the Slyne Basin record several discrete periods of growth and development. There is significant evidence for halokinesis during the Early and Middle Jurassic, with the crests of fault-blocks cored by salt rollers eroded by the base-Upper Jurassic Unconformity. There is also evidence for Permian salt diapirs forming in the Central Slyne Sub-basin during the Early to Middle Jurassic which collapsed during the Late Jurassic extensional episode, as recorded in the reduced Lower and Middle Jurassic section observed in narrow fault-bounded grabens (e.g. Fig. 9C; Vendeville & Jackson, 2001; O'Sullivan et al., 2021). Several other halokinetic structures were also reactivated during Late Jurassic extension, including the structure containing the Corrib gas field, with significant Late Jurassic throw recorded on the listric fault above the Triassic salt wall (Fig. 9B). Some of these salt structures have also undergone minor modification during the Cretaceous and Cenozoic.





## 6. Structural Evolution of the Slyne Basin

### 6.1.     Permian and Triassic

Post-Variscan extension began in the Slyne Basin during the Late Permian. Several hundred metres of Zechstein halite was deposited throughout the Slyne Basin (Fig. 9A), likely in fault-bounded depocentres (O'Sullivan et al., 2021). The Permian boundaries of the Slyne Basin are poorly understood due to post-Permian halokinesis, but it is clear that the Slyne Basin was an area of active extension, relative to the neighbouring Erris and Porcupine basins, with a thin (10s of metres thick) layer of predominately clastic and carbonate facies developed in the former (Robeson et al., 1988; O'Sullivan et al., 2021), and no evidence of Permian sediments in the latter (Jones & Underhill, 2011; Bulois et al., 2018).

The Triassic was a period of relative quiescence in the Slyne Basin, typified by the near isopachous nature of the Lower Triassic section throughout the basin (Fig. 5, 7, 9). The local thickening of the Upper Triassic section in the synclines flanking the Corrib anticline (Fig. 7, 9B) suggest that low-strain extension may have begun during the Late Triassic, at least in the Northern Slyne Sub-basin (O'Sullivan & Childs, 2021).

### 6.2.     Early and Middle Jurassic

Low-strain regional extension occurred throughout the Slyne Basin during the Early and Middle Jurassic. The Lower and Middle Jurassic section can be observed thickening towards the basin-bounding faults in the Central Slyne Sub-basin by a few 10s of ms TWTT (10-100 metres), but this shape is accentuated by erosion of this section at the Base Upper Jurassic Unconformity on the basin margins (e.g. well 27/5-1 location in Fig. 5). The Lower and Middle Jurassic section is also observed thickening into the synclines flanking the salt-cored folds in the Northern Slyne Sub-basin (Fig. 7, 9B), indicating the Permian salt was undergoing halokinesis during this period (O'Sullivan & Childs, 2021). There is also evidence of salt walls forming in the Central Slyne Sub-basin during the Early to Middle Jurassic along with large salt rollers beneath active listric faults soling out in the Permian Zechstein Group (Fig. 9C, D). In the Northern Slyne Sub-basin, a comparison of the stratigraphic section encountered in basinward wells with the single available well located on the footwall of the basin-bounding faults demonstrates the growth in the Lower and Middle Jurassic section during this period of regional extension (Fig. 10).




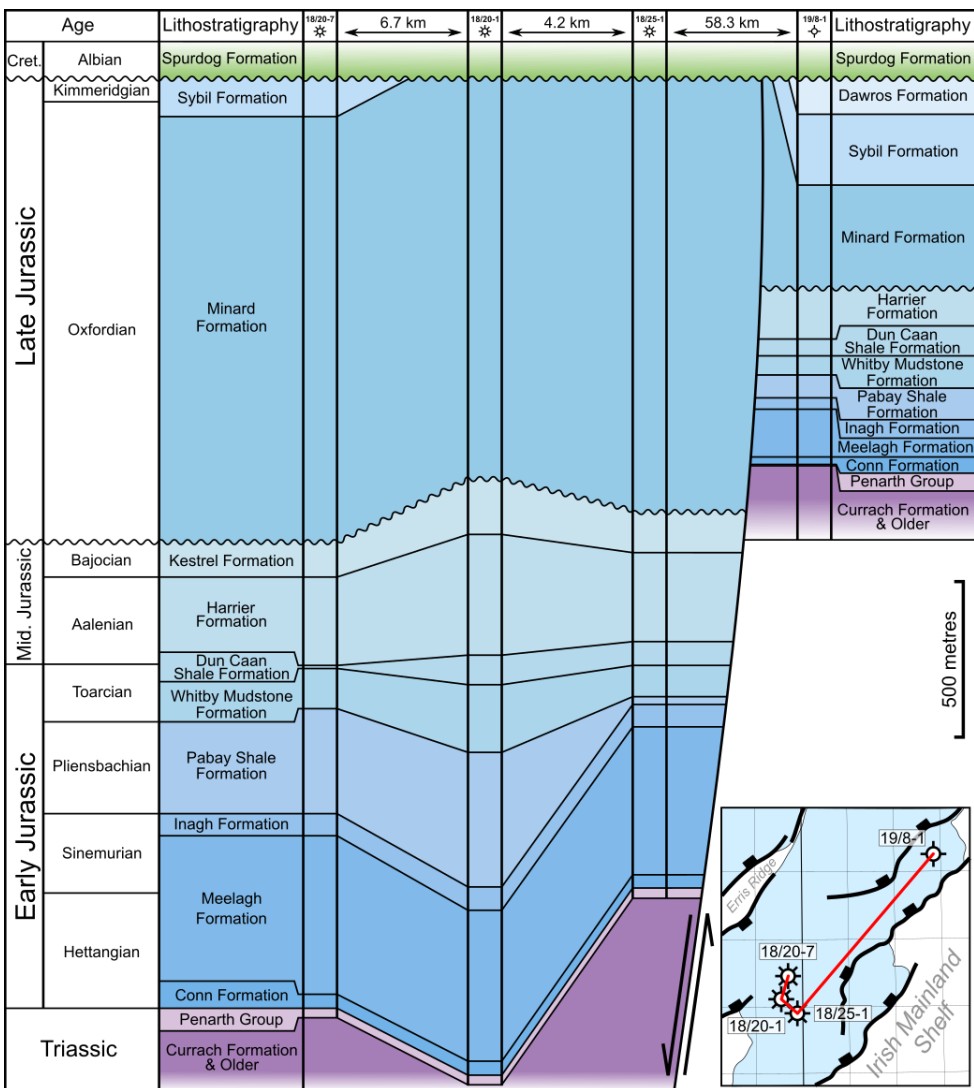

**Figure 10:** Well correlation through the Jurassic section of key wells from the Northern Slyne
Sub-basin, highlighting thickness variations in the Lower and Middle Jurassic section between
wells within the basin and the 19/8-1 well on the footwall of the basin-bounding fault system.

A regional unconformity separates the Lower to Middle Jurassic from the Upper Jurassic
section throughout the Slyne Basin, termed the Base Upper Jurassic Unconformity. This
unconformity can be quite rugose on the margins of the basins, such as the area around the
27/5-1 well in the Central Slyne Sub-basin (Fig. 5), while being a relatively flat paraconformity
in the centre of the basin (e.g. Fig. 5, 7). There are several angular truncations observed
throughout the Slyne Basin at the base of this unconformity, particularly above salt-related
structures formed during Early to Middle Jurassic extension, including footwalls above salt



rollers and the crests of folds above salt pillows (Fig. 9C, D). Throughout the Slyne Basin the
late Middle Jurassic (Bathonian and Callovian) section is absent at this unconformity, either
through erosion or non-deposition (Merlin Energy Resources Consortium, 2020). The exact
cause of this unconformity is difficult to constrain, although some authors have suggested
thermal doming and dynamic topography above a mantle plume similar to that implicated in
the North Sea (Tate & Dobson, 1989; Underhill & Partington, 1993; Doré et al., 1999).

## 6.3.  Late Jurassic

The main phase of extension commenced during the Late Jurassic, with the basin-bounding
faults accumulating several kilometres of throw during this extensional episode along with the
deposition of several kilometres of Upper Jurassic sediment (Fig. 4-8). Despite this, there are
no obvious growth sequences observed in the Southern Slyne Sub-basin (Fig. 4) or in the
southern portion of the Northern Slyne Sub-basin (Fig. 6). Growth sequences are observed in
the hanging walls of the bounding faults in the Northern Slyne Sub-basin with reflectors
diverging towards the SE (Fig. 7). In the Central Slyne Sub-basin, the Upper Jurassic section
onlaps the flank of the high-relief monocline in the immediate hanging wall of the basin-
bounding fault and thickens into the hanging wall of major intra-basinal listric fault (Fig. 5).
This stratal geometry, along with a similar thickness of Lower-Middle Jurassic sediment
present in the neighbouring Slyne Embayment, suggests that most of the throw on this fault
accumulated during the Late Jurassic, with the kilometre-scale post-rift uplift and erosion
during the Cretaceous and Cenozoic removing any Jurassic sediment from the intervening
footwall, forming the Slyne High (Fig. 5). The presence of NE-SW oriented fault splays in the
sub-salt hanging wall of this fault (e.g. Fig. 11) suggests that the large NNE-SSW oriented
fault bounding the Central and Southern Slyne Sub-basins formed through the linkage of NE-
SW oriented fault segments, likely during this Late Jurassic phase of rifting.





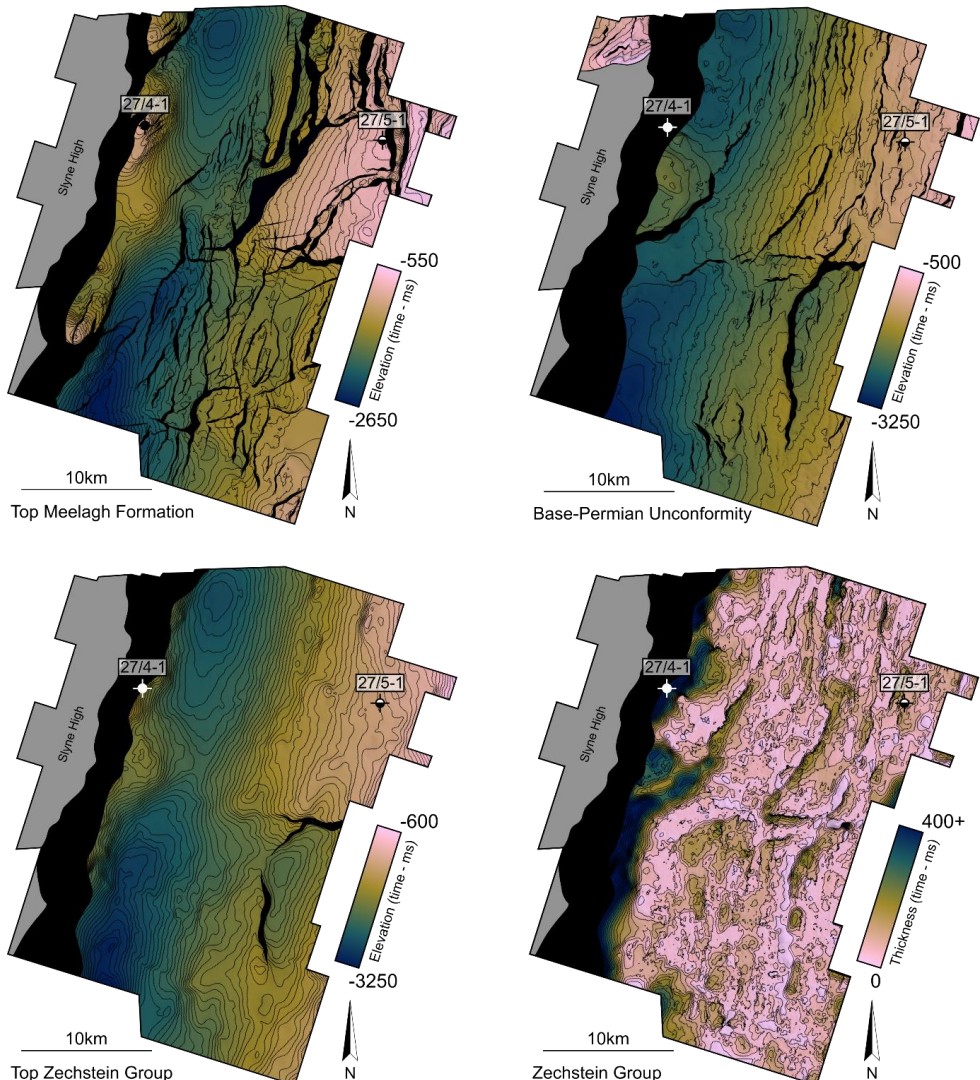

**Figure 11:** Surfaces from the E00IE09 3D seismic volume from the Central sub-basin **A)**
TWTT structure map of the Top Meelagh Formation. Several high-relief anticlinal closures are
present in the immediate hanging-wall of the basin-bounding fault, including the structure
containing the 27/4-1 'Bandon' oil accumulation. **B)** TWTT structure map of the Variscan
Unconformity. Notice the significant differences in fault pattern between the Variscan
Unconformity (pre-salt) and Top Meelagh Formation (post-salt). **C)** TWTT structure map of the
Top Zechstein Group. Notice the lack of faulting on this surface. **D)** TWTT thickness map
(isochron) of the Zechstein Group. The Zechstein salt is thinned throughout most of the survey
area, with numerous apparent welds formed between the post- and pre-salt sections. The
Zechstein salt is overthickened in the immediate hanging wall of the basin-bounding fault.

Two discrete phases of Late Jurassic extension have been identified in the neighbouring

Porcupine Basin, the first occurring in the Oxfordian and the second in the Kimmeridgian





(Saqab et al., 2020). Both of these extensional episodes may have also occurred in the Slyne
Basin but, unlike the Porcupine Basin, a significant section of the Late Jurassic syn-rift section
was subsequently removed during post-rift uplift and erosion (e.g. Corcoran & Mecklenburgh,
2005) and evidence of a second phase may have been removed.

## 427 6.4.    Cretaceous and Cenozoic

The Slyne Basin experienced kilometre-scale uplift and erosion at the end of the Jurassic and
during the Early Cretaceous, removing a significant section of the Upper Jurassic syn-rift
section throughout the basin (Table 1). The majority of the Slyne Basin was likely a
topographic high relative to surrounding regions during the Cretaceous, including the Erris,
Porcupine and Rockall basins (Fig. 8; Musgrove & Mitchener, 1996; Chapman et al., 1999;
Saqab et al., 2020). Up to 400 metres of Albian and Late Cretaceous sediments were
deposited in the Northern Slyne Sub-basin and the Slyne Embayment (5-8, 12B), and possibly
in the Central and Southern Slyne sub-basins. Several syn-rift faults were reactivated during
the Cretaceous, with both normal and reverse movement observed throughout the Slyne
Basin. In the Northern Slyne Sub-basin the main delamination fault above the Corrib anticline
has a significant Cretaceous growth sequence that thickens from 200 ms TWTT (c. 150 m) in
the footwall to over 400 ms TWTT (c. 380 m) in the hanging-wall (Fig. 7). Additionally, the
individual segments of the basin-bounding fault system along the eastern margin of the
Northern Slyne Sub-basin were reactivated during the Cretaceous (Fig. 12B). The throw on
the northern segment varies from 30-100 ms TWTT adjacent to the Corrib gas field through to
the 19/8-1 well (Fig. 7, 8) while on the segment to the south adjacent to the 18/25-2 well (Fig.
6) the throw locally exceeds 300 ms TWTT. In addition to these major faults, several smaller
faults offset the Cretaceous section throughout the Northern Slyne Sub-basin with the majority
of these faults having throws less than 100 ms TWTT (Fig. 6, 7). The fault bounding the Slyne
Embayment appears not to have been active during this time, with Cretaceous sediments
overstepping the fault with no offset (Fig. 5). The absence of Cretaceous sediments in the
Central and Southern Slyne sub-basins obscures any fault activity that may have occurred
during this period (Fig. 12B). Nevertheless, due to the pervasive nature of Cretaceous faulting
in the Northern Slyne Sub-basin and strong evidence of Cretaceous faulting in the Porcupine
Basin to the southwest (Jones & Underhill, 2011; Saqab et al., 2020), it is likely that some
structures in the Central and Southern Slyne sub-basins were active during the Cretaceous.
The motion on these faults would likely have been less than 100 ms TWTT in a similar manner
to those in the Northern Slyne Sub-basin. Alongside the reactivation of Jurassic syn-rift faults,
the majority of which were oriented NNE-SSW parallel to the axis of the Slyne Basin, a new




set of ENE-WSW oriented faults formed during the Cretaceous, observed offsetting the upper
100-200 ms TWTT of the Upper Jurassic section and the Cretaceous section in the Northern
Slyne Sub-basin (O'Sullivan & Childs, 2021).

| Exhumation estimate (Km) | Location | Source |
|---|---|---|
| 0.7-1.9 | 27/13-1 | Scotchman & Thomas, 1995 |
| 0.8-1.7 | Corrib | Corcoran & Mecklenburgh, 2005 |
| 0.7-3.2 | Central and Southern Slyne sub-basins | Biancotto & Hardy, 2007 |
| 1.6-2.0 | 27/24-sb02 | Fugro, 1994b |
| 1.8 | 27/5-1 | Geotrack, 1996 |
| 0.8-2.6 | 19/8-1 | Geotrack, 2008 |

**Table 1:** Exhumation estimates from different locations throughout the Slyne Basin. Well
locations are shown in Figures 1 and 3.

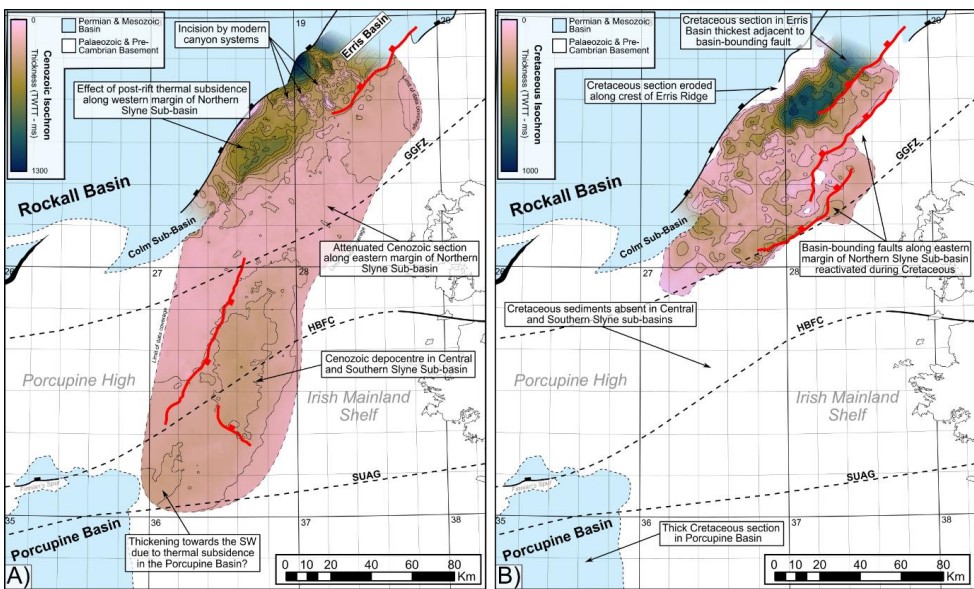

**Figure 12: A)** TWTT thickness map (isochron) of the Cenozoic section in the Slyne and
southern Erris Basins superimposed on the main syn-rift structural features. A thicker Cenzoic
section is observed along the margin of the Rockall Basin on the western margin of the
Northern sub-basin and in the southern Erris Basin. This is transected by modern slope
canyons which incise into the Cenozoic section. A thicker Cenozoic section is also observed
in the Central sub-basin. **B)** TWTT thickness map (isochron) for the Cretaceous section in the
Slyne and southern Erris Basins superimposed on the main syn-rift structural features.
Cretaceous strata are absent in the Slyne Basin south of the Central Slyne Transfer Zone but
is present in the North Porcupine Basin. A significantly thicker Cretaceous section is preserved
in the southern Erris Basin, although it is eroded along the north-western margin of the Erris
Basin.





A second period of uplift and erosion occurred during the early Cenozoic throughout the Slyne
Basin, forming another regional unconformity (Fig. 4-8). This was accompanied by a period of
regional magmatism, expressed as igneous intrusions observed throughout the Slyne Basin
(Fig. 4, 6, 7, 8) and layers of basaltic lava in the Northern and Southern Slyne sub-basins (Fig.
4, 7).

Cenozoic tectonic activity reactivated several structures throughout the Slyne Basin with
different expressions and senses of motion in different sub-basins; In the Northern Slyne Sub-
basin the delamination fault above the Corrib anticline was reactivated for a second time,
offsetting the early Eocene lavas of the Druid Formation, alongside the large listric fault to the
west of Corrib (Fig. 7). In the Central and Southern Slyne sub-basins, several intra-basinal
faults were reactivated, with both normal and reverse motion observed on faults with Cenozoic
throw between 10 to 50 ms TWTT (Fig. 5). The large listric fault in the Central Slyne-Sub basin
was inverted along with some of the rafted fault blocks along the eastern margin of the basin
(Fig. 5). In the Central Slyne Sub-basin the bounding fault along the western margin of the
basin was reactived during the Cenozoic, with between 50-150 ms TWTT of throw recorded
along its length (Fig. 5, 12A). The faults bounding the Northern Slyne Sub-basin were not
reactivated during the Cenozoic (Fig. 6-8, 12A) but due to thermal subsidence in the
neighbouring Rockall Basin the Cenozoic sequence thickens significantly along the western
margin of the Northern Slyne Sub-basin (Fig. 7, 8, 12A).
## 7. Discussion
### 7.1.   Structural inheritance and the impact of oblique pre-
existing structures

Structural inheritance is a common feature across the sedimentary basins of NW Europe, with
Carboniferous, Permian, Jurassic and Cretaceous rifting interpreted to reactive older, pre-
existing structures which formed during the Caledonian or Variscan Orogenies. This is
recorded along the Atlantic margin of NW Europe (Stein, 1988; Doré et al., 1999; Ziegler &
Dèzes, 2006; Schiffer et al., 2019) as well as in the basins of the North Sea (Fazlikhani et al.,
2017; Philips et al., 2019; Osagiede et al., 2020). The reactivation of structures has been
observed both onshore and offshore Ireland, with faults in the Carboniferous basins in the Irish
midlands forming parallel to the NE-SW structures in the Caledonian basement (Worthington
& Walsh, 2011; Kyne et al., 2019), while Variscan structures form the template for the later
development of the Celtic Sea basins (Van Hoorn, 1987; Shannon, 1991; Rodriguez-Salgado
et al., 2019). Similar relationships have been suggested for the Irish Atlantic margin (Tate &





Dobson, 1989; Naylor & Shannon, 2005), with several Caledonian structures mapped onshore
continuing into the offshore domain (Fig. 1).
The relationship between pre-existing basement structure and basin formation has been
studied extensively from outcrop and subsurface mapping and using analogue modelling (e.g.
Tommasi & Vauchez, 2001; Fazlikhani et al., 2017; Collanega et al., 2019). The key factor
that determines the nature of this relationship is the relative orientation of inherited structure
and the later extension direction (e.g. Henza et al., 2011; Henstra et al., 2015). Where
inherited structures are at a low angle to the extension direction, they are not reactivated but
may impede the propagation of new extensional faults and may give rise to transfer zones
between adjacent fault/basin segments. As the angle between pre-existing structures and
extension direction increases the likelihood of reactivation of basement structure increases
and analogue modelling has demonstrated the variety of fault patterns that can form in the
cover sequence. Although the effect of basement structure can be manifest in many ways the
two situations that have received most attention are extension oblique to an individual
basement fault (Schlishe et al., 2002) and oblique basin opening modelled by extension
oblique to a zone of weakness (Agostini et al., 2009; Philipon & Corti, 2016). In both cases
extension results in the formation of new fault segments, or faults, that are normal, or close to
normal, to the extension direction and arranged en echelon above or within the basement
structure or zone. Figure 13B illustrates fault/basin geometry that is characteristic of extension
oblique to a basement fabric; the key feature is that the overall orientation of the structure is
parallel to the basement structure. The Slyne Basin does not follow Caledonian basement
structure but cuts across it and as a result displays a different style of inheritance. Figure 13A
illustrates our interpretation of the initial Jurassic geometry of the Slyne Basin that is based on
observations below; this geometry resembles that in Fig. 13C in which individual fault
segments follow the basement trend but the basin as a whole cuts across it.

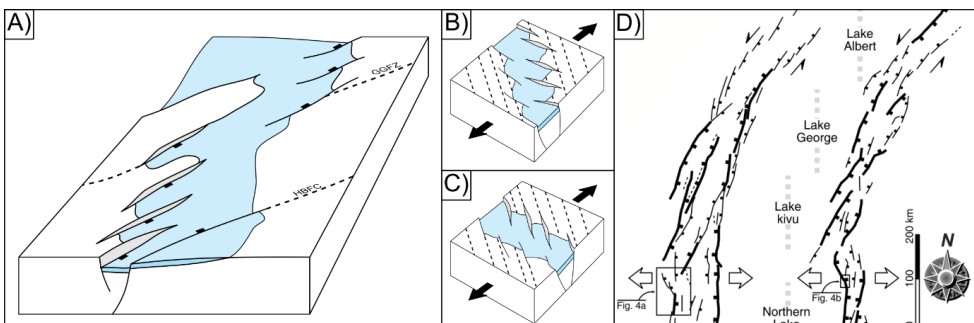


**Figure 13: A)** Schematic block model showing the initial fault segments of the basin bounding
faults and the reversal in basin polarity across the GGFZ. **B-C)** Blocks models showing
different patterns of basement formation when extension is oblique to pre-existing structures.





**D)** Section of Figure 3 of Corti et al. (2007) demonstrating similar rift geometries to those observed in the Slyne Basin.

The Slyne Basin strikes NNE-SSW (020°) and cuts across the local Caledonian inherited trend oriented NE-SW (c. 045°). On the eastern flank of the Northern Slyne Sub-basin the bounding faults parallel the Caledonian trend and form a left stepping fault array (Fig. 1). The map pattern on the western flank is somewhat obscured by erosion and data quality but the faults also parallel the Caledonian trend. Within the Central Slyne Basin the faults offsetting the Jurassic are predominantly parallel to the basin axis (Fig. 11A). The majority of these faults are confined to the Jurassic section and are decoupled from the Carboniferous basement by the Zechstein salt (Fig. 5). The fault forming the western flank of the Central Slyne Basin is approximately parallel to the basin trend (Fig. 1) but closer inspection (Fig. 11A) shows that it has a distinct splay in the sub-salt basement. This fault pattern is consistent with this margin of the basin originating as a left-stepping fault array that would have comprised fault segments parallel to the preserved splays i.e. at a strike of ca. 040° and close to the orientation of the Caledonian basement fabric. We suggest therefore that the main faults that bound the Slyne Basin during Jurassic extension initially comprised left stepping arrays of fault segments that individually followed the Caledonian NE-SW trend (Fig. 13A, 14). This initial segmentation is preserved in the fault array bounding the eastern margin of the North Slyne Sub-basin but was bypassed by the formation of a through-going, basin-parallel (i.e. NNE-SSW oriented) fault in the Central Slyne Sub-basin. One of the main Caledonian structures that transects the basin, the Great Glen Fault Zone, was one of the structures reactivated to form a segment of the eastern margin of the North Slyne Basin and also perhaps one of the segments of the western margin of the Central Slyne Basin (the bounding fault of the Slyne Embayment), acted as the zone across which the basin reversed polarity.





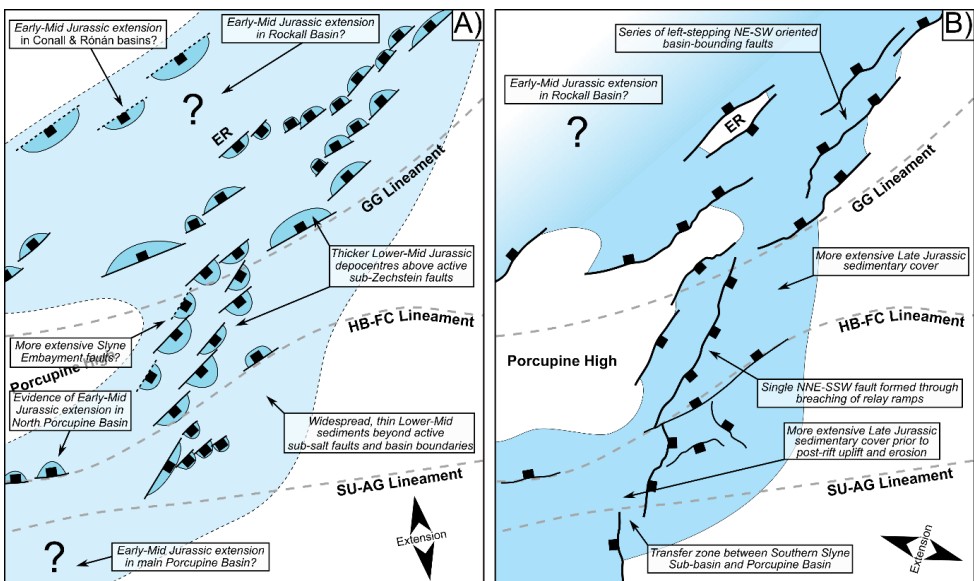

**Figure 14:** Conceptual maps showing the evolution of the main basin-bounding and intra-basinal faults in the Slyne Basin and surrounding areas during the **A)** Early to Middle Jurassic and **B)** Late Jurassic.

The Slyne Basin provides an example of a form of basement control that is not frequently documented in the literature. In general, individual faults are sub-perpendicular to the extension direction, whether they be segments of a fault located above a reactivated basement structure or faults within an oblique rift. The Slyne Basin cuts across the basement trend but the individual basin bounding faults follow the basement trend. These two styles of interaction are compared in Figure 13B and 13C. A key difference between these is that there is a reversal of the sense of stepping of the basin bounding faults despite the fact that the angular relationship between the basement and the extension direction is the same. This style of inheritance is not generally recognised in analogue modelling, but Corti et al. (2007) generated this pattern by introducing discrete narrow zones of weakness (Fig. 13D). Their model, designed to replicate the structure of the western branch of the East African Rift System, generated left-stepping rift-bounding faults in the presence of E-W extension by reactivation of discrete crustal structures in a pattern very similar to that seen in the Slyne Basin. While this style of inheritance is perhaps unusual, there are other areas in which it can be observed. In the northern North Sea, a Triassic-Jurassic broadly N-S rift system formed on crust with both N-S and NE-SW oriented Devonian and Caledonian crustal structures display a wide variety of styles of inheritance (Fazilkhani et al. 2017; Phillips et al. 2019) but a common feature is that major faults that parallel Caledonian trends are left stepping and the map pattern of the Viking Graben, for example, is similar to that of the East African Rift shown in Fig.13D.





## 7.2. Post-rift uplift and erosion

A significant section of the syn-rift section is absent from the Slyne Basin, with key structural
geometries recorded in the Upper Jurassic syn-rift sequences missing due to kilometre-scale
uplift and erosion during the Cretaceous and Cenozoic (Table 1). The magnitude of uplift and
erosion throughout the Slyne and Erris basins is highly variable. Previous authors have
recorded a wide range of values for the magnitude of this post-rift exhumation, ranging from a
few hundred metres to several kilometres (Scotchman & Thomas, 1995; Corcoran & Clayton,
2001; Doré et al., 2002; Corcoran & Mecklenburgh, 2005; Biancotto et al., 2007). This
variability in exhumation estimates arises due to the geological complexity associated with this
process; in the Slyne Basin, three discrete post-rift unconformities are observed: the Base-
Cretaceous, Base-Cenozoic and mid-Miocene unconformities. These unconformities are the
result a variety of local and regional tectonic processes, including rift-shoulder uplift associated
with rifting and hyperextension in the neighbouring Rockall Basin, the development of the
Icelandic plume and the North Atlantic Igneous Province, ridge-push at the Mid-Atlantic Ridge,
the Alpine Orogeny, and possibly the development of the Brendan Igneous Centre (Fig. 1, 2;
Mohr, 1982). Additionally, these unconformities become composite surfaces at different
locations within the Slyne basin: the absence of Cretaceous strata in the Central and Southern
Slyne Sub-basins (Fig. 12B) may be due to non-deposition, or more likely, the erosion of the
thin Cretaceous section, similar to that observed in the Northern Slyne Sub-basin, during the
Cenozoic uplift events. The formation of these composite unconformities obscures the
reactivation of any syn-rift faults in the Central and Southern Slyne Sub-basins during the
Cretaceous, as the circa 100-300 m of erosion at the Base-Cenozoic Unconformity (sensu
Corcoran & Mecklenburgh, 2005) is greater than the throw recorded on most of the
Cretaceous faults observed in the Northern Slyne Sub-basin (Fig. 6, 7). Finally, the multitude
of methodologies used to estimate exhumation varies throughout the basin, and includes
vitrinite reflectance (Scotchman & Thomas, 1995; Corcoran & Clayton, 2001), compaction
analysis (Corcoran & Mecklenburgh, 2005), and analysis of seismic velocities (Biancotto et
al., 2007).
A further consequence of this extensive and variable erosion during the Cretaceous and
Cenozoic is that the present-day boundaries of the basin are not representative of their full
extent during the main syn-rift period. The Mesozoic rift basins on the Irish Atlantic margin,
including the Slyne Basin, were much more extensive prior to uplift and erosion. Consequently,
some publications have, as a result, focused on individual basins as separate and different
geological entities rather than as residual parts of a complex, margin-wide rift system. Possible
reconstructions of the Slyne Basin and neighbouring areas during the Early-Middle and Late
Jurassic periods are presented in Figure 14.





## 7.3. The Slyne Basin in the context of the Irish Atlantic margin

As stated above, the Slyne Basin belongs to a framework of basins which stretch across the Irish Atlantic margin and likely shares aspects of its geological evolution with these other areas. The most similar of these neighbours is the Erris Basin directly north of the Northern Slyne Sub-basin (Fig. 1). The Erris Basin is contiguous with and has a similar sedimentary fill to the Slyne Basin which suggest that both basins underwent a similar geological evolution during the Permian, Triassic and Jurassic periods (Fig. 8). The evolution of the Slyne and Erris basins diverges in the Cretaceous, with the thicker Cretaceous section in the Erris Basin (Fig. 8, 12B) indicating it underwent active extension during the Cretaceous alongside the neighbouring Rockall Basin while the Slyne Basin remained largely inactive.

The Slyne Basin is separated from the Porcupine Basin by a narrow basement high approximately five kilometres wide (Fig. 4). This high is the eroded footwall of the fault bounding the Southern Slyne Sub-basin, with kilometre-scale erosion largely taking place during the Cretaceous and Cenozoic (Dancer et al., 1999; Biancotto et al, 2007). Restoring a kilometre-scale section of Upper Jurassic stratigraphy would connect the Porcupine Basin with the Southern Slyne Sub-basin, supporting the idea that these basins developed coevally during the Late Jurassic (Fig. 14B). The nearby 26/30-1 well in the Porcupine Basin (Fig. 4) encountered the Upper Jurassic Minard Formation resting unconformably atop the Carboniferous Blackthorn Group (Phillips Petroleum Company, 1982), while the intervening Permian to Middle Jurassic stratigraphy present in the Southern Slyne Sub-basin is absent. While Triassic and Lower Jurassic stratigraphy has been encountered in two wells in the North Porcupine Basin to the north of the Finnian's Spur (Fig. 1B; Bulois et al., 2018; Merlin Energy Resources Consortium, 2020), most wells in the Northern Porcupine Basin encountered Upper Jurassic sediments resting directly atop Carboniferous sediments (Merlin Energy Resources Consortium, 2020). Permian sediments have not been encountered in any well in the Porcupine Basin (Merlin Energy Resources Consortium, 2020). This indicates that the Slyne Basin is the older of the two basins, with extension beginning in the Late Permian with the deposition of several 100 metres of Zechstein Group evaporites (Štolfová & Shannon, 2009; O'Sullivan et al., 2021) while the Northern Porcupine likely remained a relative high during the latest Palaeozoic and early Mesozoic. There may be narrow outliers of Permian, Triassic and Early to Middle Jurassic-aged sediments preserved beneath the Late Jurassic sediments further south in the Porcupine Basin, but at present this remains unproven.



## 8. Conclusions

Detailed interpretation of available seismic reflection data in conjunction with borehole and potential-field datasets has delivered an improved understanding of the complex and multiphase structural history of the Slyne Basin.

1. The onset of rifting in the Slyne Basin began in the Late Permian, expressed as diffuse extensional faulting accompanied by the deposition of the Zechstein Group evaporites in localised, fault-bounded depocentres. This was followed by tectonic quesence during the majority of the Triassic and subsequent extension accompanied by localised halokinesis during the Latest Triassic and into the Early and Middle Jurassic. Regional uplift and erosion occurred during the late Middle Jurassic, creating a regional unconformity. The main phase of rifting began in the Oxfordian and continued until the end of the Jurassic.

2. The Slyne Basin experienced kilometre-scale uplift and erosion throughout the Early Cretaceous, creating the distinct angular unconformity between Jurassic syn-rift sediments and Cretaceous and younger post-rift sediments. Subsequent and less-severe phases of exhumation occurred during the Cenozoic. Faults throughout the basin are reactivated in both normal and reverse senses during this tectonic activity.

3. The segmentation of the Slyne Basin into discrete sub-basins occurs where crustal-scale structural lineaments, representing the suture zones and boundaries between Caledonian and Precambrian terranes, obliquely transect the younger Mesozoic basin.

4. The basin axis is oriented NNE-SSW and cuts across the N-E Caledonian trend resulting in a rarely documented style of fault reactivation in which the segments of basin-bounding faults follow the earlier structural grain but the basin as a whole does not. As strain increased initial left-stepping segments linked resulting in basin-bounding faults oriented parallel to the basin axis.

5. Salt layers in the Slyne Basin exert important controls on basin-development, most importantly acting as décollements between the Palaeozoic pre-salt basement and Mesozoic post-salt basin-fill. The most important salt-prone interval is the Permian Zechstein Group, present throughout the Slyne Basin, while in the Northern sub-basin the Upper Triassic Uilleann Halite Member is also present, acting as a second layer of mechanical detachment.

## 9. Data availability

The data that support the findings of this study were provided by the Petroleum Affairs Division (PAD) and are available for download from https://www.dccae.gov.ie/en-ie/natural-resources/topics/Oil-Gas-Exploration-Production/data/Pages/Data.aspx. Restrictions may apply to the availability of these data, which were used under licence for this study.



## 10. Author contribution

Conor O'Sullivan carried out data analysis, wrote the original text, drafted the figures, and conceptualised the original ideas presented therein. Conrad Childs and Mudasar Saqab provided initial project conceptualisation, supervision and reviewed the final text. John Walsh and Patrick Shannon reviewed the final text.

## 11. Declaration of competing interests

The authors declare that they have no known competing financial interests or personal relationships that could have appeared to influence the work reported in this paper.

## 12. Acknowledgements

This research is funded in part by a research grant from Science Foundation Ireland (SFI) under Grant Number 13/RC/2092 and is co-funded under the European Regional Development Fund, and by the Petroleum Infrastructure Programme (PIP) and its member companies. Efstratios Delogkos is thanked for thoughtful discussions regarding the links between the Slyne Basin and the Bróna and Pádraig basins. Karize Oudit, Neil Jones, Blanca Cantalejo Lopez and Andrew King of CNOOC International are thanked for engaging discussions on the structural evolution of the Slyne Basin. Phil Copestake of Merlin Energy Resources Limited is thanked for providing additional detail on revised biostratigraphic interpretation of the Slyne Basin. The authors would like to thank the Petroleum Affairs Division (PAD) of the Department of Communications, Climate Action and Environment (DCCAE), Ireland, for providing access to released well, seismic and potential field datasets. Europa Oil & Gas are thanked for providing access to the Inishkea 2018 reprocessed 3D volume and allowing an arbitrary line from the volume to be shown. Shell Exploration & Production Ireland Ltd. are thanked for providing access to reprocessed volumes of the 1997 Corrib 3D. The authors would also like to thank Schlumberger for providing academic licenses of Petrel to University College Dublin.



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
