# Peer review of "Tectonostratigraphic evolution of the Slyne Basin"

_EGUsphere, 2022_

## Referee Comment (RC1)

[referee-annotated manuscript omitted]

---

## Author Comment (AC3)

| Reviwer | Original Line | Reviewer comment | Agree | Response comment |
|---|---|---|---|---|
| 1 | Throughout | Grammar and spelling corrections throughout the manuscript text included in attachment | Accepted | These corrections have been made throughout the text. |
| 1 | Abstract | The abstract refers to two tectonic events, but only one post-rift unconformity. Please, check this | Accepted | Refined wording to indicate the three rift phases and the presence of multiple post-rift unconformities |
| 1 | Introduction | There are published examples in the literature of Paleozoic fault zones controlling the formation of rift basins in deep offshore Portugal. The most documented of such examples is the Messejana Fault Zone in SW Portugal - see Pereira and Alves (2013). Crustal deformation and submarine canyon incision in a Meso-Cenozoic first-order transfer zone (SW Iberia, North Atlantic Ocean). Tectonophysics 601, 148-162. | Accepted | included reference to Pereira & Alves 2013 in introduction when outlining types of rift segmentation and structural inheritance. |
| 1 | Figures | Provide/make map of Caledonain suture zones | Accepted | Defined terrane types (taken from Stolfova & Shannon, 2007) on Fig. 1B and added additional labels to the suture lines to make them clearer |
| 1 | Figures | Update tectonic event column | Responded | To avoid confusion, the events column has been removed from Fig. 2 and a simple seismic horizon column has been added (see reviewer 2 comment below) |
| 1 | 168-170 | Update ages and causes of key sequences in relation to the ages stated in Figure 2 events column. In this case the Bay of Biscay rifting and formation of oceanic crust. | Responded | While the authors agree that far-field events are impacting the evolution of the Slyne Basin, the strongest driver during the Cretaceous period is the rifting and hyperextension in the Rockall Basin immediately to the NW. Reference has been added other far-field events, including rifting in the Bay of Biscay at this time. |
| 1 | Salt section | Quantify the role of salt in basin evolution and preservation of the petroleum system. | Accepted | The Slyne Basin has undergone relatively little shortening (less than 1% in total) with all syn-rift structures still having significant net-normal offsets recorded. The scale of salt-related fault movement has been added to the end of the Role of Salt section to quantify this. As this paper is not dealing with the petroleum system specifically, it was chosen to exclude any mention to it beyond mentioning relevant structures such as the Corrib gas field, to avoid making the scope to wide or the text to convoluted. |
| 2 | 14 | I'd add what age is "Caledonian". It would really help phrase it in the context | Accepted | Reword to include the Silurian-Devonian age of the Caledonian orogeny |
| 2 | 15 | I'd potentially add "Initial" at the beginning of this sentence. It helps a nonexpert understand they should expect a sequential sequence of rifting. | Accepted | Added intially at start of sentence |
| 2 | 35-36 | You are referring here to Variscian orogenic and the opening of the Atlantic Ocean, but Figure 1A legend says Palaeozoic & Pre-Cambrian basement. I think that it would make it easier to add an age to Variscian orogeny in the text, even if this is still early in the introduction. This comment will also help understand the relationship between the Caledonian and Variscian orogenies in the next sentence. | Accepted | Added ages for both Variscan orogeny and oceanic crust formation to bookend the geological periods being discussed here. |
| 2 | 39 | The transition here is not easy. Are you referring to the Caledonian or Variscan structures? I'd maybe write something like "Later rift event had either reactivated Caledonian/Variscan age structure if oriented optimally (REFs) or were segmented, hindering fault growth, in cases rift structures were oblique (REFs)". | Accepted | Added mention of both orogenies to the final sentence in this paragraph to link it with arguments presented in prior sentences |
| 2 | 55 | I would suggest adding: "The Slyne basin (XX km long and YY km wide)…" | Accepted | Added basin measurements to this sentence |
| 2 | 58 | You make it sounds like transfer zones are specific description of these structures located in the Slye basin. I think that you could make it a little more coherent by: "In the case of the Slye basin, these transfer zones have been…" | Accepted | Reworded this sentence to not suggest these zones are specific to the slyne basin. |

| Reviewer | Line | Comment | Status | Response |
|---|---|---|---|---|
| 2 | 60-62 | I'm sure I miss understand something here, but it reads like you're suggesting that transfer zones are areas of reactivation of pre-existing zones of weakness. To my understanding, transfer zone area areas where normal faults transfer strain (Morley et al. 1990; Childs et al. 2019). Could you make it clearer what is your definition of transfer zones? | Accepted | Reworded to say that pre-existing structures can localise where transfer zones form during fault propagation. |
| 2 | 92-93 | Can you highlight the location of the Caledonian terrane boundaries are in Figure 1. | Accepted | Added sentence at end of paragraph highligthing the terrane boundaries are in Fig. 1B |
| 2 | 94-102 | Very clear description to a great figure. | Accepted | Thanks! |
| 2 | 112 | I'd add a reference to Figure 1B at the end of this sentence. I might also think if there is a way to somehow highlight in that great figure what are the Caledonian structures? It would really help differentiating between them and the younger faults. | Accepted | Included reference to Fig. 1B. In the authors opinion, the location of Caledonian structures are clear in this figure, given their location on the otherwise white basement area around the basement and different line style. Label size has been increased and addtional labels have been added to the west of the basin to make these structures clearer. |
| 2 | 149 | I'd change to: "The Carboniferous mudstones are overlain..." just to make it clear you are not referring to the Silurian metasediments you ended the sentence with. | Accepted | Added mention of Carboniferous to clarify what the Permian sediments overlie |
| 2 | 200 | The horizons are not in Figure 2. It would be great to have them in. | Accepted | Added type seismic section to Fig. 2 |
| 2 | 260 | please add reference to Figure 1B at the end of this. This is a very cool observation, but complicated to understand without the figures in hand, so reference to figures is helpful. | Accepted | |
| 2 | 273-274 | Not sure that I completely agree. You said that the exact location of the GGFZ is not that clear in that area, I don't think you can conclude that it acted as a barrier to the propagation of the Slyne Embayment basin bounding fault. | Accepted | Reworded to say that the deformation in the basement associated with the GGFZ acted as a preferential area to transfer strain between the younger basin-bounding faults. |
| 2 | 274-275 | I'd just add why you think they are linked. I guess you think that because of their geographical location? If so, you could add something like: "As the GGFZ transect the CSTZ and is located between the fault bounding the Slyne Embayment and the southernmost segment of fault system bounding the Northern Slyne Sub-basin, there is probable structural link between these fault systems." | Accepted | As above, changed wording to indicate pre-existing deformation helped localise strain transfer between the younger faults across this area. |
| 2 | 280-283 | A bit long and convoluted. Could you split into two sentences, stopping after the reference to the figure. Does the HBFC has a sinistral component? If so, I think it might be helpful to add that in the text and add strike-slip arrows to figure 1B (and also to the other basement faults). | Accepted | Split sentences and added more detail into the description of the map-view pattern and fault offset. |
| 2 | 295-296 | Figure 2 shows two small rectangles indicating Triassic salt in the southern basin, so maybe worthwhile removing them from Figure 2. A reference to Figure 9A could also be useful here. | Accepted | Added refernce to Figure 9A in-text and edited Figure 2 |
| 2 | 333-335 | could you please add reference to Figure/other paper like you did in the next sentence? As I would guess that most readers of this manuscript are not salt-tectonic experts, it would really help demonstrate your point. | Accepted | Added reference to figure 9C and D which both show tilted beds folded during Early-Middle Jurassic salt movement subquently eroded during the late Middle Jurassic |
| 2 | Chapter 6 | The transition between the results (Chapter 5) and Chapter 6 makes the reading a little difficult as it is not clear what is the difference between Chapter 6 (Structural Evolution of the Slyne Basin) and Chapter 3 (Geological Settings). This is because Chapter 6 is followed by a Discussion chapter (Chapter 7). I think could easily be fixed with a sentence or two describing the role of Chapter 6 (Maybe worth thinking about making it a sub-chapter within the Discussion) | Accepted | Changed section numbering so that section 6 is now 5.3, making it a sub-heading of the results. A short preamble to introduce this section has also been added. |

| | | | Comment | Status | Response |
|---|---|---|---|---|---|

| 2 | Chapter 6 | | I think that adding thickness maps for each unit will be very useful to illustrate the relationship between the different structural elements. These cross-sections are helping, and are doing a good job at making the point you are trying to make, but as you have such an abundance of seismic data it would be really cool to see that. I understand this would require a significant amount of work, so I leave that to the authors discretion | Responded | Great suggestion. Unfortunately the author no longer has access to the data to create regional thickness maps. |
| 2 | Chapter 6 | | Great Glen is strike slip. Does the HBFC and the SUAG are also strike slip? If so, do they have the same direction? If no what is their original offset and how would you think that influence the location of the later faults? Does transtension/transoression have influenced the location of the basin-bounding faults or the geometry of the Slyne basin? | Accepted | Added more detail in the Basement overview section to describe the shape of these structures and elements of minor strike-slip movement observed onshore Ireland. |
| 2 | Chapter 6 | | Adding a block diagram/simplified map similar to Figure 14 which demonstrates the evolution which you can use refer back throughout Chapter 6 would be useful. As there almost no use of the 3D volumes, I think that as the paper is titled tecono-stratigraphy of the basin, this sort of figure would be very helpful to a nonexpert reader. | Responded | Great suggestion. A similar figure was drafted prior to initial submission but due to the structural change observed along-strike it was ultimately not included due to it being more confusing than helpful. The authors believe this is still the case and map-view evolutionary diagrams are more useful in this instance. |
| 2 | 361-364 | | I think that either a thickness map or illustration on the crosssection is needed here. It's very hard to judge based on the cross-sections provided if the thickness was salt-tectonic related or related to other extension related structures occurring at the time. | Responded | Great suggestion. Unfortunately the author no longer has access to the data to create regional thickness maps. |
| 2 | 394-396 | | This is an interesting and non-trivial observation! At first, I was certain the Late Jurassic had growth strata, but I totally agree. Does that mean this thick unit is pre-kinematic to the main extension phase? These thickness changes are not trivial, and I would suggest adding some indication for this in the cross-sections to help readers understand that. I'd also think that a little more details on why you think that growth strata are present in the Central and Northern Slyne basins, but not in the Northern basin. Or how these faults in the Northern basin had accumulated such thick strata with not apparent growth at all. Sounds like a key feature not only to the understanding of the basin development, but also to the strain interaction between the two sub-basins! | Accepted | Added indication to the thickness changes in Fig. 5 and they are already highlighted in Fig. 7. Added more detail in-text describing how these relate to the evolution of the Slyne and Porcupine basins at the end of the Late Jurassic section. |
| 2 | | 402 | "this fault", does that mean the basin-bounding fault or the intra-basinal listric fault? | Accepted | Clarified that this refers to the fault bounding the Central Slyne Sub-basin |
| 2 | 435-437 | | Not clear what syn-rift episode you are referring to. A reference to a figure is missing here. | Accepted | Clarified this refers to faults active during the Late Jurassic |
| 2 | 513-519 | | A simplified figure is missing here to help explain this text. I think you can add more text/details to Figure 13 B&C to show how the angle between extension and pre-existing structure will affect the resulted structures. Also, as this is a critical text to your assumptions, if you could add a reference here it would help (maybe to the analogue modelling you wrote). | Accepted | The figure caption for Figure 13 B&C has been updated to match the in-text description and improve the overall message of the figure. This should more clearly illustrate the two phenomena being described here. |

| | | | | |
|---|---|---|---|---|
| 2 | 525-531 | I couldn't follow the differences between Figure 13B&C, or fully understand the text. An example for the confusion is: "… characteristic of extension oblique to the basement fabric; the key feature is that the overall orientation of the structure is parallel to the basement structure". It's not very clear who are the structures, what is the basement fabric and what are the basement structures, mainly because it's not very clear from Figure 13B what you are referring to in the text. Amending the text or adding arrows/text in the figures could help. | Accepted | Reworded this section and changed the use of the more general 'structures' to instead relate to the basin, basement structures and faults. Updated the figure caption to describe the differences between both block diagrams. |
| 2 | 542 | an earlier explanation on what is the Caledonian trend (potentially in the Geological settings) would have made it much easier to understand. See my previous comments. | Responded | The Caledonian trend is already highlighted in the Basement Configuration section in the Geological Setting |
| 2 | 552-555 | Could you suggest why the initial segmentation is preserved in the Northern Slyne basin but not in the southern? Is it the activation of the Great Glen? | Accepted | Added two speculative mechanisms for the different evolution in the basin-bounding fault systems |
| 2 | General | There are multiple syn-rift episodes, it would be easier if you could number/give age throughout the text when you refer to them. It's not easy to follow 'main syn-rift' or 'syn-rift', especially when the rifting episodes are not coeval in all sub-basins | Accepted | Removed any mention of 'main syn-rift' and clarified these events as belonging to the Late Jurassic syn-rift phase. |

---

## Author Response (AR1)

Re: Manuscript Revision

Dear Editor,

My co-authors and I thank you for securing two excellent reviewers for our manuscript. We are very appreciative of the feedback our manuscript has received from both. We have found this feedback both positive and constructive and feel it has greatly refined our original submission. Please see below a summary of the changes we have made to the manuscript, alongside an attached table outlining the specific responses to reviewer comments in detail.

In general, we have:

- Improved detail, clarity and information on all figures.
- Revised the text to remove typos and make arguments more focused.
- Refined the terminology when referring to particular tectonic events to avoid confusion.
- Added more quantification regarding post-rift salt-related fault movement.
- Added more detail, both in-text and in figures, to describe the basement structure in the study area.

With these revisions in mind, we hope this manuscript has suitably addressed the shortcomings of our previous submission and satisfies the reviewers' comments. The co-authors and I are still confident this manuscript will be an important contribution to the understanding of salt tectonics and salt distribution on the European Atlantic margin. We look forward to hearing from you and please do not hesitate to contact me if you require any additional material or clarification.

Yours sincerely,

Dr. Conor O'Sullivan (on behalf of the co-authors)

Structural Geologist

Petroleum Experts Ltd.,
Petex House,
10 Logie Mill,
Edinburgh,
EH7 4NJ,
United Kingdom.
Telephone: +44 (0) 7479343956 / +353 (0) 851441181
Email: cmnosullivan@gmail.com